# High Expression of HERV-K (HML-2) Might Stimulate Interferon in COVID-19 Patients

**DOI:** 10.3390/v14050996

**Published:** 2022-05-07

**Authors:** Yaolin Guo, Caiqin Yang, Yongjian Liu, Tianyi Li, Hanping Li, Jingwan Han, Lei Jia, Xiaolin Wang, Bohan Zhang, Jingyun Li, Lin Li

**Affiliations:** Department of AIDS Research, State Key Laboratory of Pathogen and Biosecurity, Beijing Institute of Microbiology and Epidemiology, Beijing 100071, China; guoyaolin1997@outlook.com (Y.G.); y_17311983041@163.com (C.Y.); yongjian325@sina.com (Y.L.); litianyi1983@yahoo.com (T.L.); hanpingline@163.com (H.L.); hanjingwan@outlook.com (J.H.); jialeihaowawa@gmail.com (L.J.); woodsxl@163.com (X.W.); zbhforjob@163.com (B.Z.); lijyjk@163.com (J.L.)

**Keywords:** SARS-CoV-2, COVID-19, interferon, HERV-K (HML-2), Gene Ontology

## Abstract

**Background.** Interferon is a marker of host antiviral immunity, which is disordered in COVID-19 patients. ERV can affect the secretion of interferon through the cGAS-STING pathway. In this study, we explored whether IFN-I and HERV-K (HML-2) were activated in COVID-19 patients and whether there was an interaction between them. **Methods.** We collected blood samples from COVID-19 patients and healthy controls. We first detected the expression of HERV-K (HML-2) *gag*, *env*, and *pol* genes and IFN-I-related genes between patients and healthy people by qPCR, synchronously detected VERO cells infected with SARS-CoV-2. Then, the chromosome distributions of highly expressed HERV-K (HML-2) *gag*, *env*, and *pol* genes were mapped by the next-generation sequencing results, and GO analysis was performed on the related genes. **Results.** We found that the HERV-K (HML-2) *gag*, *env*, and *pol* genes were highly expressed in COVID-19 patients and VERO cells infected with SARS-CoV-2. The interferon-related genes *IFNB1*, *ISG15,* and *IFIT1* were also activated in COVID-19 patients, and GO analysis showed that HERV-K (HML-2) can regulate the secretion of interferon. **Conclusions.** The high expression of HERV-K (HML-2) might activate the increase of interferon in COVID-19 patients, proving that HERV-K does not only play a negative role in the human body.

## 1. Introduction

Severe acute respiratory syndrome coronavirus 2 (SARS-CoV-2), the seventh coronavirus to infect humans, has been raging around the world since the end of 2019 [1,2]. Most patients with COVID-19 show mild or moderate symptoms, such as fever and cough. The disease progression of severe patients is often more serious and can progress to acute respiratory distress syndrome (ARDS), pneumonia, acute heart injury, multiple organ failure, and secondary infection [3,4,5,6,7,8]. Irreversible pulmonary fibrosis is also present in cured patients, and potential damage to the reproductive and haematopoietic systems can occur [9]. These injuries are associated with the development of uncontrolled systemic inflammation in the organism following infection.

As an important effector molecule of the innate immune signalling pathway, interferon (IFN) is closely related to cytokine storms in patients with COVID-19, and it is also a hallmark of host antiviral immunity. Interferon can be divided into three types [10]. Type I interferon (IFN-I) includes IFN-α, -β, -ε, -κ, and -ω in humans, type II interferon (IFN-II) has only IFN-γ, and type III interferon (IFN-III) includes IL-29 (IFN-λ1), IL-28A (IFN-λ2) and IL-28B (IFN-λ3). Secreted IFN binds the receptor IFNAR to the cell surface and then activates Janus-activated kinase-signal transducers and activators of transcription (JAK-STAT1) to induce the expression of IFN-stimulated gene (ISG) and interferon-induced proteins with tetratricopeptide repeats (IFIT) [11,12]. Working together with ISGs, IFIT family proteins are involved in many processes in response to viral infection, mainly by reducing virus replication [13,14]. This disruption of the intrinsic immune system is concerning, as we found in our previous work that interferon was significantly elevated in patients with moderate or severe COVID-19 [15,16,17,18].

A study in *Cell* showed that endogenous retrovirus (ERV) activates the cGAS-STING pathway in mice to affect the secretion of interferon [19]. Human endogenous retroviruses (HERVs) are ancient integrations of exogenous viruses. Through Mendelian inheritance and evolutionary processes spanning millions of years, they now occupy approximately 8% of the human genome [20]. HERV-K (HML-2) is the most active transcriptional subtype recently acquired and plays a critical role in embryogenesis [21,22,23]. HERV-K is silenced in most cell types in healthy adults, and studies have shown that its reactivation is associated with a variety of cancers [24,25,26]. As the main functional genes, HERV-K *gag*, *env* and *pol* gene encode matrix protein, envelope and polymerase, respectively [27]. High expression of HERVs was also found in the lavage fluid of COVID-19 patients [28], but the activation of HERV-K in patients with different clinical types was not reported. In view of the above phenomena, we suspected that HERV-K (HML-2) has an important ability to activate interferon, thus we explored whether the activation of interferon in moderate COVID-19 patients and severe COVID-19 patients is related to HERV-K (HML-2).

In this study, COVID-19 patients were divided into two groups: moderate and severe. We first detected the activation of HERV-K (HML-2) and the expression of interferon-related genes, then mapped the chromosome distribution of highly expressed HERV-K (HML-2) *gag*, *env*, and *pol* genes, and finally analysed the related genes by GO. The overall results confirmed our conjecture that the activation of HERV-K (HML-2) in COVID-19 patients is related to the increase in interferon, thus providing a new idea for the mechanism of interferon production in COVID-19 patients.

## 2. Materials and Methods

### 2.1. Patients and Healthy Donors

Sixty-four SARS-CoV-2-positive individuals (COVID-19) and thirty-two healthy individuals were enrolled from Beijing Youan Hospital. The study was approved by the Beijing Youan Hospital Research Ethics Committee (No. 2020-037, 15 April 2020). Written informed consent was obtained from all participants According to the Chinese Government Diagnosis and Treatment Guidelines (8th edition), the clinical classification was divided into moderate and severe patients. To inactivate the whole blood sample complement, all samples were incubated in a water bath at 56 °C for 30 min.

### 2.2. Virus Infection

Vero E6 cells were cultured in Dulbecco’s modified Eagle medium (DMEM, Gibco, New York, NY, USA) supplemented with 10% foetal bovine serum (FBS) and 1% penicillin–streptomycin (Gibco, New York, NY, USA). To prepare the virus stock, 5 × 10^6^ Vero E6 cells were infected with SARS-CoV-2 (BetaCoV/Beijing/AMMS01/2020) at a multiplicity of infection (MOI) of 0.01. After culturing for 48 h, the cells were incubated in a water bath at 56 °C for 30 min. The cells in the culture flask were completely lysed with 6 mL TRIzol. The cell control procedure was the same as above and without infection with SARS-CoV-2. All experiments with the SARS-CoV-2 virus were conducted in the BSL-3 laboratory.

### 2.3. Total RNA Extraction

Total RNA was extracted from whole blood using the automatic nucleic acid separation and purification system (MagNA Pure LC 2.0 Roche, Basel, Switzerland) following the RNA Isolation Kit-High Performance (REF.03542394001) without modification. RNA samples were evaluated by NanoPhotometer-N60 (Implen, München, Germany), showing a ratio of 260/280 of approximately 2.0 and a concentration ranging from 10 ng/mL to 100 ng/mL.

### 2.4. Reverse Transcription

Using a HIFIScript gDNA Removal cDNA Synthesis Kit (NO. CW2582 M, Cowin Bio, Suzhou, China), each sample was first placed in a PCR amplification instrument (Bio–Rad T100, Hercules, CA, USA) according to the proportion of 1 µL 10 × gDNA eraser buffer, 0.5 µL gDNA eraser, and 8.5 µL RNA template (total 2.5 ng) and incubated at 42 °C for 5 min to remove gDNA. Then, 10 µL gDNA-free reaction solution, 1 µL HIFIScript, 1 µL primer mix, 4 µL 5 × ScriptRT buffer, and 4 µL RNase-free water were placed in a PCR amplification instrument (Bio–Rad T100, Hercules, CA, USA) and incubated at 42 °C for 30 min followed by 85 °C for 5 min. Finally, cDNA was obtained.

### 2.5. Quantitative Reverse Transcription PCR (qRT–PCR)

qRT–PCR was performed with MagicSYBR Mixture (NO. CW3008H, Cowin Bio, Suzhou, China) using a Roche LightCycler 480 II System. The amplifications were run in a 96-well plate at 95 °C for 10 min, followed by 40 cycles of 95 °C for 5 s, 60 °C for 30 s, and 72 °C for 10 s, and then a melting curve analysis was performed to confirm the specificity of the PCR. Each sample was run in triplicate. Relative quantification of target gene transcripts was performed with the ∆Ct method, and the ∆Ct value was corrected by the corresponding *β-actin* (HERV-K-related gene) or *GAPDH* (IFN-β-related gene) control Ct values. The primer sequences are provided in Table 1.

### 2.6. Genome-Wide Distribution Map

The samples of COVID-19 patients were amplified by PCR with the same qPCR primers, and the products were purified and sent for next-generation sequencing (SinoGenoMax, Beijing, China). The sequencing results were aligned to the reference genome (GRCh38/hg38) on UCSC (www.genome.ucsc.edu, accessed on 10 December 2021). The chromosomal loci with identity values greater than 97% were screened, and the gene expression rates of these loci were counted by uniqueseqs values. Then, we used the R packet *RIdeogram* to construct the distribution location of HERV-K (HML-2) *gag*, *env*, and *pol* high expression fragments on chromosomes and provide a visual heatmap for tracking labels [29,30,31].

### 2.7. Functional Annotation and Classification

To identify functional categories of highly expressed genes, Gene Ontology (GO) [32] enrichment analysis was performed using the GREAT Input: Genomic Regions Enrichment of Annotations Tool (http://great.stanford.edu/public/html/, accessed on 29 December 2021).

### 2.8. Statistical Analysis

All continuous variables are described as medians (IQRs), and categorical characteristics are described as numbers (%). Means for continuous variables were compared using independent group t tests when the data were normally distributed; otherwise, the Mann–Whitney test was used. All statistical analyses were performed using SPSS (Statistical Package for the Social Sciences) version 21.0 software (SPSS Inc, CA, USA). The values * *p* < 0.05, ** *p* < 0.01, and *** *p* < 0.001 were considered significant; ns stands for not significant.

## 3. Results

### 3.1. Demographic Characteristics

The demographic characteristics of 72 COVID-19 patients are summarized in Table 2. This study included 49 moderate and 23 severe COVID-19 patients. There were 42 females and 30 males; the median age was 63.8 years, and 72.22% of patients were over the age of 60. We enumerated the PCR-positive detection time of moderate and severe patients and found that there was no difference in PCR-positive detection time between the two groups by the Mann–Whitney test. The average ages of moderate and severe patients were similar; 63.6 and 64.3, respectively.

### 3.2. SARS-CoV-2 Infection Increased the Expression of HERV-K (HML-2) gag, env, and pol

Previous studies have shown that interferon production is related to ERV in mice [19], and we detected the expression of HERV-K (HML-2) *gag*, *env*, and *pol* genes in the whole blood of COVID-19 patients and VERO cells infected with SARS-CoV-2 (Figure 1). The expression of HERV-K (HML-2) *gag*, *env*, and *pol* genes was increased in moderate and severe COVID-19 patients (*p* < 0.05), but there were no differences between the two clinical types.

In VERO cells infected with SARS-CoV-2, we found that the expression of HERV-K (HML-2) *env* and *pol* genes was significantly increased (*p* < 0.05), while the HERV-K (HML-2) *gag* gene expression was increased, but not significantly (*p* > 0.05).

### 3.3. SARS-CoV-2 Infection Increased the Expression of Interferon-Related Genes

To explore the mechanism of the increase in interferon, we detected the expression of genes related to interferon expression, including *IFNB1* (interferon beta 1), *ISGs*, and *IFITs*. Expression of the *IFNB1* and *IFIT1* genes in moderate COVID-19 and severe COVID-19 patients was higher than that in healthy people (*p* < 0.05), but there was no difference between the two clinical types, which was consistent with that of serum IFN-γ. There was no difference in the expression of the *ISG15* gene among the three groups (Figure 2).

In VERO cells, we found that the expression of the *IFIT1* gene decreased in VERO cells infected with SARS-CoV-2 (*p* < 0.05), there was no difference in the expression of the *ISG15* gene between infected and uninfected VERO cells, and the expression of the *IFNB1* gene could not be detected in either group.

### 3.4. Chromosomal Location of Highly Expressed HERV-K (HML-2) gag, env, pol

After next-generation sequencing and screening of chromosome loci with identity greater than 97%, we found that 25 HERV-K (HML-2) *gag*, 36 *env* and 37 *pol* loci were highly expressed in the human genome (Figure 3, Appendix A). The highest expression site of *gag* was on chr 12 (approximately 58,335,037–58,335,251), and the lowest expression site was on chr 1 (approximately 75,378,750–75,378,940). The highest expression site of *env* was on chr 3 (approximately 101,699,823–101,699,990), and the lowest expression site was on chr 19 (approximately 22,580,569–22,580,735). The highest expression site of *pol* was on chr 7 (approximately 4,586,046–4,586,188), and the lowest expression site was on chr 1 (approximately 13,212,726–13,212,855).

### 3.5. Gene Ontology (GO) Classifications

GO analysis provides a description of gene products in terms of their associated biological process (BP), cellular component (CC), and molecular function (MF) [33,34]. Typical enriched GO terms are shown in Figure 4 (Appendix A). Regarding HERV-K (HML-2) *gag*, a total of 34 unigenes were categorized into 39 GO terms, including 12 biological processes, 17 cellular components, and 10 molecular function terms. Biological regulation (GO:0065007), membrane (GO:0016020), and protein binding (GO:0005488) were the most highly represented GO terms in BP, CC, and MF, respectively. Regarding HERV-K (HML-2) *env*, a total of 48 unigenes were categorized into 42 GO terms, including 12 biological processes, 18 cellular components, and 12 molecular function terms. Biological regulation (GO:0065007), membrane (GO:0016020), and protein binding (GO:0005488) were the most highly represented GO terms in BP, CC, and MF, respectively. Regarding HERV-K (HML-2) *pol*, a total of 51 unigenes were categorized into 42 GO terms, including 12 biological processes, 18 cellular components, and 12 molecular function terms. Biological regulation (GO:0065007), membrane (GO:0016020), and protein binding (GO:0005488) were the most highly represented GO terms in BP, CC, and MF, respectively.

### 3.6. “Interferon Gamma Signalling” Was Highly Enriched According to KEGG Mapping

The Kyoto Encyclopedia of Genes and Genomes (KEGG) database is a collection of various pathways representing molecular interactions and reaction networks [35]. To clarify the signal transduction pathway of the gene at the chromosome site where HERV-K (HML-2) *gag*, *env*, and *pol* were highly expressed in COVID-19 patients, we mapped the KEGG database and found that the identified differentially expressed genes (DEGs) were significantly enriched in 30 KEGG pathways (*p* value ≤ 0.05, Appendix A). The top 10 enriched pathways of HERV-K (HML-2) *gag*, *env*, and *pol* genes are shown in Figure 5. DEGs were highly clustered in several signalling pathways, such as “SUMO is proteolytically processed”, “Ligand–receptor interactions”, “Beta defensins”, “Defensins”, and “Netrin-1 signalling”, suggesting that HERV-K (HML-2) *gag*, *env*, and *pol* genes may perform their functions through these pathways. Notably, “interferon gamma signalling” was identified as the second most significantly enriched pathway in KEGG analysis (enrichment ratio = 8.0898 and *p* value < 0.05).

## 4. Discussion

A central paradigm of immunity is that IFN-mediated antiviral responses precede proinflammatory responses, optimizing host protection and minimizing collateral damage [36,37]. Some studies have shown that IFN-γ is significantly increased in COVID-19 patients, and the antiviral activity of IFN-I, IFN-II, and IFN-III counterbalances ACE2 inducibility and restricts SARS-CoV-2 [38]. Although coronavirus can evade and inhibit the IFN response to achieve high pathogenicity [39,40,41,42,43], much evidence has shown that IFN-I and IFN-III still have the potential to treat SARS, MERS, and COVID-19 [44,45,46,47,48,49,50]. As a HERV family closely related to the growth and development of the host, HERV-K is often related to the abnormal activation of diseases, and many biomarkers of disease take this as the development point [26,51,52]. In this study, we confirmed that the activation of HERV-K leads to the activation of the IFN pathway, which provides a new direction for the potential mechanism of interferon production; at the same time, it also provides a new view of the physiological function of HERV-K. We think that when disease occurs, HERV-K participates in the regulation of immunity and accelerates the recovery from disease.

Our study first noticed a significant increase in serum IFN-γ in moderate and severe COVID-19 patients, and IFN-γ with antiviral ability was also characterized to be associated with the titre of the neutralizing antibody in COVID-19 patients [53]. IFN-γ participates in the cytokine storm and can interact with TNF-α to activate inflammatory CXCL10+ CCL2+ macrophage subsets and cause inflammatory cell death, tissue damage, and mortality via SARS-CoV-2 infection [10,54]. In addition, IFN-γ is a double-edged sword; a large number of researchers have found that IFN-γ can be used as an effective method to treat cytokine storms in COVID-19 patients, and the treatment effect is better in severe patients [55]. IFN-I has a strong ability to interfere with virus replication, and recombinant IFN-β has the potential to reduce the disease risk of severe COVID-19 patients [56,57]. Our study found that the high expression of IFN-β in COVID-19 patients induced an increase in *ISG15* and a high expression of *IFIT1* (*ISG56*), the main subclass of ISG proteins with broad-spectrum antiviral activity, which may be the antiviral immune mechanism induced by SARS-CoV-2 infection [13,58,59].

VERO cells are derived from African green monkey kidney and act as hosts for ERV and SARS-CoV-2, which is a good model for this study. VERO cells are also widely used in the study of the molecular mechanism of virus infection, as well as the production of vaccines and recombinant proteins, and are regarded as an ideal cell model for cultivating influenza vaccines and studying the molecular mechanism of virus infection [60,61,62]. VERO cells could not secrete interferon α/β when infected with a virus because of their inherent genetic defects. Barrett tried to infect VERO cells with Newcastle disease virus, Sendai virus, and rubella virus to produce interferon, while the results showed that VERO cells could not express the antiviral protein interferon [63]. Interestingly, our study found that no significant changes in *IFNB1* or *ISG15* were detected in VERO cells infected with SARS-CoV-2. However, the expression of *IFIT1*, a negative feedback regulator that dampens virus-induced innate immune signalling, decreased significantly [64]. Even if SARS-CoV-2 can antagonize interferon, we suspect that the vaccine produced by a cell line that secretes interferon in its entirety would be more effective. However, it is difficult to find a cell line that can not only adapt to viral infection but also secrete interferon by itself, and this conjecture needs to be confirmed by much research.

Djalma S. used mouse subjects and found that cDNA formed by reverse transcription of highly expressed ERVs after microbial infection of the skin activates the cGAS-STING pathway and induces the production of IFN-I [19]. HERVs are also aberrantly activated in many diseases. Here, we selected the HERV-K (HML-2) family, which is closely related to human development, to explore whether SARS-CoV-2 could also activate HERV-K (HML-2) through upper respiratory tract infection. Significantly high expressions of the HERV-K (HML-2) *gag*, *env*, and *pol* genes were detected in both moderate and severe COVID-19 patients. The gene expression of HERV-K (HML-2) is under the direct control of its long terminal repeats (LTRs), studies have shown that the HIV-1 *tat* gene can induce transcriptional activation of HERV-K(HML-2) LTR [65,66,67]. We speculate that the high expression of HERV-K (HML-2) *gag*, *env*, and *pol* genes in whole blood of COVID-19 patients may also be due to the similar activation of some specific SARS-CoV-2 genes, which increase the expression of the LTR region. While HERVs were randomly inserted into the human genome, we further identified the chromosome site where the highly expressed HERV-K (HML-2) *gag*, *env*, and *pol* genes are located. Through GO analysis of the possible signalling pathways regulated by these sites, it was found that HERV-K (HML-2) *gag* was related to the interferon signalling pathway, and Pearson correlation analysis showed that HERV-K (HML-2) *gag* was positively correlated with *IFNB1*, *ISG15*, and *IFIT1* (Appendix A, *p* < 0.05), which perfectly confirmed the hypothesis that HERV-K (HML-2) activated interferon secretion in COVID-19 patients. Unfortunately, we did not have enough whole blood samples from COVID-19 patients to verify this finding at the protein level, which leads to a limitation of our study.

In our GO analysis, we found that the HERV-K (HML-2) *gag*, *env*, and *pol* genes all regulated the SUMO proteolytic process. SUMOylation is a reversible posttranslational modification that distinctly regulates IFN synthesis, IFN signalling and the expression and function of IFN-stimulated gene (*ISG*) products and plays a key role in the interferon pathway and antiviral defence [68,69]. Therefore, HERV-K (HML-2) may not only activate the cGAS-STING signal but also perhaps control interferon secretion by regulating the SUMO pathway. In addition, previous studies have found that IFN-λ1 IFN-I and beta defensins are upregulated in COVID-19 patients over 15 years of age. Our GO analysis results show that HERV-K (HML-2) is also closely related to beta defensins. High expression of HERV-K (HML-2) may affect the secretion of beta defensins and promote immune defence [70,71].

We suggest that the role of HERV-K in regulating human diseases may be bidirectional. Previous studies have found that HERV-K is activated in a variety of cancers, and HERV-K *env* antagonizes the antiviral activity of Tetherin, so most scholars believe that the significance of HERVs is negative [72,73]. Research on HERV-K and disease tends to take HERV-K as a biomarker of disease. In contrast, we believe that HERV-K can also play a positive role in the human body. Our research strongly supports our idea that HERV-K (HML-2) activation in COVID-19 patients might promote cGAS-STING pathway activation and regulate the SUMO pathway and β-defensin production, thus inducing IFN-I production.

## Figures and Tables

**Figure 1 viruses-14-00996-f001:**
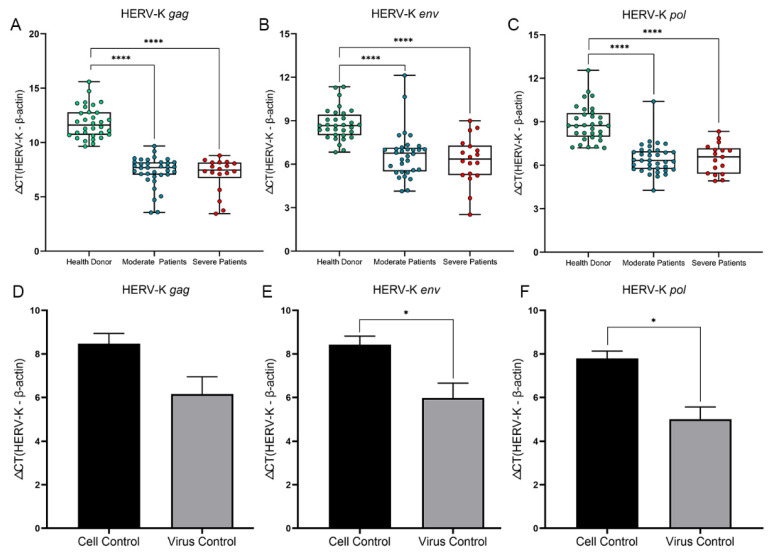
Comparison of the HERV-K (HML-2) *gag*, *env*, and *pol* genes in COVID-19 patients and VERO cells infected with SARS-CoV-2. (**A**–**C**) The expression of HERV-K (HML-2) *gag*, *env*, and *pol* genes in health donor (green dot), moderate (blue dot) and severe COVID-19 patients (red dot). (**D**–**F**) The expression of HERV-K (HML-2) *gag*, *env*, and *pol* genes in VERO cells control (black column) and VERO cell infected with SARS-CoV-2 (grey column). Each point represents three repeats. The box plots show the medians (middle line) and first and third quartiles (boxes), and the whiskers show the minimum and maximum. Mann–Whitney test *p* values are depicted in the plots. Mann-Whitney test P-values are depicted in the plots. * *p* < 0.05, **** *p* < 0.0001.

**Figure 2 viruses-14-00996-f002:**
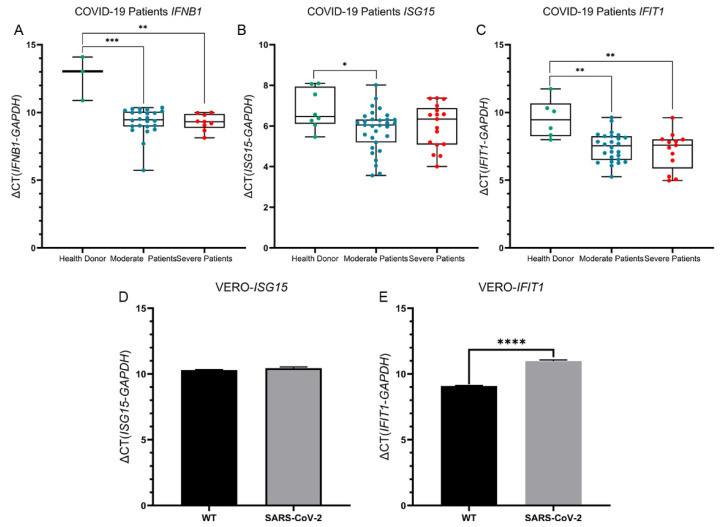
Comparison of interferon-related genes in COVID-19 patients and VERO cells infected with SARS-CoV-2. (**A**–**C**) The expression of *IFNB1*, *ISG15*, and *IFIT1* in health donor (green dot), moderate (blue dot) and severe COVID-19 patients (red dot). (**D**–**E**) The expression of *ISG15* and *IFIT1* in VERO cells control (black column) and VERO cell infected with SARS-CoV-2 (grey column). Each point represents three repeats. The box plots show the medians (middle line) and first and third quartiles (boxes), and the whiskers show the minimum and maximum. Mann–Whitney test *p* values are depicted in the plots. Mann-Whitney test P-values are depicted in the plots. * *p* < 0.05, ** *p* < 0.01, *** *p* < 0.001, **** *p* < 0.0001.

**Figure 3 viruses-14-00996-f003:**
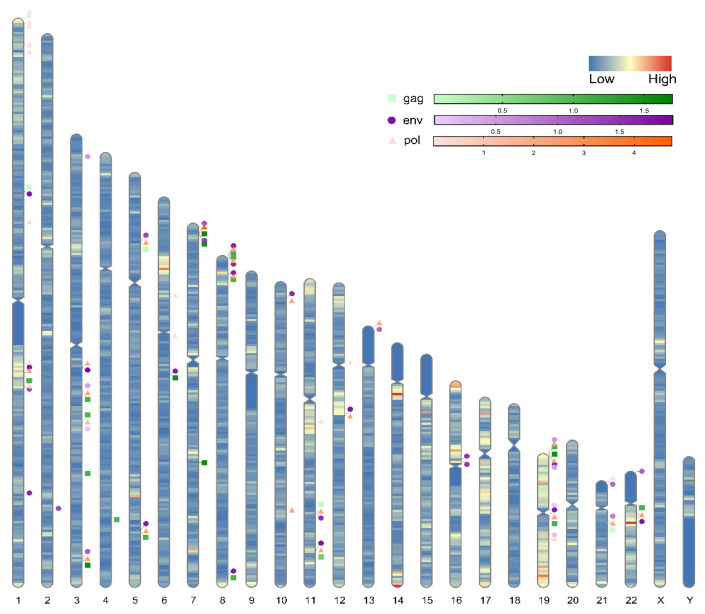
Diagram of the genome-wide distribution of highly expressed HERV-K (HML-2) *gag*, *env*, and *pol*. There are 25 HERV-K (HML-2) *gag*, 36 *env* and 37 *pol* loci in the human genome that are highly expressed. The *gag* locus is represented by a green square, the *env* locus is represented by a purple circle, the *pol* locus is represented by an orange triangle, and the gradual colour represents the relative gene expression at different loci.

**Figure 4 viruses-14-00996-f004:**
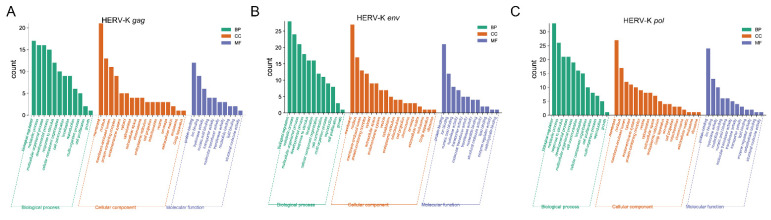
GO functional classification of differentially expressed genes (DEGs) among different samples. The distributions of highly expressed (**A**) HERV-K *gag*, (**B**) HERV-K *env* and (**C**) HERV-K *pol* are summarized in three main categories: biological process (BP), molecular function (MF), and cellular component (CC).

**Figure 5 viruses-14-00996-f005:**
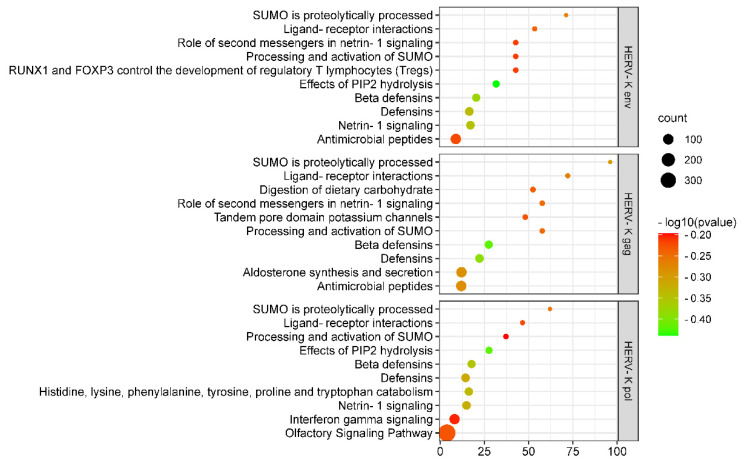
Scatter plot of enriched KEGG pathway statistics. The colour and size of the dots represent the range of the Q−value and the number of DEGs mapped to the indicated pathways, respectively. The top 10 enriched pathways are shown in the figure.

**Table 1 viruses-14-00996-t001:** Primers used in this study.

Gene Name	Direction	Primer Sequence (5′-3′)
*β-actin*-F	Forward	CCACGAAACTACGTTCAACTCC
*β-actin*-R	Reverse	GTGATCTCCTTCTGCATCCTGT
HERV-K (HML-2) *env*-F	Forward	CTGAGGCAATTGCAGGAGTT
HERV-K (HML-2) *env*-R	Reverse	GCTGTCTCTTCGGAGCTGTT
HERV-K (HML-2) *pol*-F	Forward	TCACATGGAAACAGGCAAAA
HERV-K (HML-2) *pol*-R	Reverse	AGGTACATGCGTGACATCCA
HERV-K (HML-2) *gag*-F	Forward	AGCAGGTCAGGTGCCTGTAACATT
HERV-K (HML-2) *gag*-R	Reverse	TGGTGCCGTAGGATTAAGTCTCCT
*GAPDH*-F	Forward	GGCATGGACTGTGGTCATGAG
*GAPDH*-R	Reverse	TGCACCACCAACTGCTTAGC
*IFNβ*-F	Forward	ACGCCGCATTGACCATCTAT
*IFNβ*-R	Reverse	TAGCCAGGAGGTTCTCAACA
*ISG15*-F	Forward	GAGAGGCAGCGAACTCATCT
*ISG15*-R	Reverse	CTTCAGCTCTGACACCGACA
*IFIT1*-F	Forward	TACAGCAACCATGAGTACAA
*IFIT1*-R	Reverse	TCAGGTGTTTCACATAGGC

**Table 2 viruses-14-00996-t002:** Baseline characteristics of patients infected with COVID-19.

Characteristics	Moderate Patients	Severe Patients	Total
	(*n* = 49)	(*n* = 23)	(*n* = 72)
Sex			
Female	30	12	42
Male	19	11	30
Age, y			
35–39	2	1	3
40–49	3	0	3
50–59	10	4	14
60–69	20	13	33
70–79	11	4	15
80–89	3	1	4
Median (IQR)	63.6 (58–71)	64.3 (60.5–69)	63.8 (58.8–70.2)
Days after PCR positivity			
1–7	6	3	9
8–14	11	5	16
15–21	2	2	4
22–28	14	4	18
29–35	12	7	19
36–42	3	1	4
43–49	1	1	2
Median (IQR)	22.4 (12–30)	22.6 (12–32.5)	22.5 (12–31)

## Data Availability

All data and information reported in this article will be shared by the corresponding author upon request.

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
