# Peer review of "High Expression of HERV-K (HML-2) Might Stimulate Interferon in COVID-19 Patients"

_viruses, 2022, doi:10.3390/v14050996_

Round 1
Reviewer 1 Report
This manuscript presents a potential connection between the endogenous retroviruses (ERV) with the IFN-signatures in the COVID-19 patients. Using samples from COVID-19-infected moderate and severe patients, the authors demonstrate that the ERV gene expressions were correlated with the severity of the disease. In addition, the CoV-2-infected Vero cells showed similar results in ERV expression. The infected patients showed signatures of the IFN-stimulated genes, e.g., ISG15, IFIT etc. Overall, the authors correlate the COVID-19 severity with the IFN gene signatures and ERV gene expression. The study has some novelty but also suffers from some weaknesses, as noted below
- Please use different labeling/calculations for the patient sample data in Fig 1 and 2, so the conclusion that these ERVs are increased in COVID-19 patients will be justified.
- The rationale behind the presence of ERV in Vero cells is unclear.
- What is the possible mechanism behind the increase in ERVs in COVID-19 patients? Please discuss this. How does CoV-2 increase ERV gene expression?
- Can the authors use any other virus as a negative control? How about the genes of VSV or HSV-1 in the patients? These results will reveal the specificity of the assays.
Author Response
Point 1: Please use different labeling/calculations for the patient sample data in Fig 1 and 2, so the conclusion that these ERVs are increased in COVID-19 patients will be justified.
Response: We appreciate for your nice suggestion. Figure 1 shows the expression of HERV-K (HML-2) gag, env, and pol genes in COVID-19 patients and VERO cells infected with SARS-CoV-2. Figure 2 shows the expression of interferon-related genes in COVID-19 patients and VERO cells infected with SARS-CoV-2. Figure 1 and figure 2 are both qPCR test data, so we chose to use the same label/calculations method to unify the full text of the diagram and make it easy to understand. Therefore, we believe that the same label/calculations method should be maintained here.
Point 2: The rationale behind the presence of ERV in Vero cells is unclear.
Response: Thank you for your rigorous consideration. ERV is common in many mammals, such as mice, cats, etc. Vero cells are derived from the kidney of an African green monkeys, so the ERV gene in Vero cells may also come from ancient integration. At the same time, VERO cell is a good model for studying SARS-CoV-2, so we chose this cell as control. We have also made a supplementary explanation in the line 263 on page 14.
Point 3: What is the possible mechanism behind the increase in ERVs in COVID-19 patients? Please discuss this. How does CoV-2 increase ERV gene expression?
Response: Thank you for your kind guidance. We added something about mechanism behind the increase in ERVs in COVID-19 patients in the discussion section of the article. Please see line 287 on page 15.
Point 4: Can the authors use any other virus as a negative control? How about the genes of VSV or HSV-1 in the patients? These results will reveal the specificity of the assays.
Response: We gratefully thank you for your constructive remarks. We regret that due to the limitations of sample size and revision time, we are really unable to supplement the relevant data at present. However, in our study, COVID-19 patients are all hospitalized patients, and most of the patients in this period are due to the immune response caused by SARS-CoV-2, rather than other viruses. According to the clinical data, none of the patients were infected with other viruses, so we think that the current data show that the transcriptional upregulation of HERV-K (HML-2) gag, env, and pol genes has nothing to do with other viruses.
Reviewer 2 Report
Guo et al. presented an interesting manuscript linking COVID-19 infection to high expression of human endogenous retrovirus (HERV-K) which leads to increased interferon production. This is suggesting that high expression of HERV-K is not only leading to pathological conditions (as for example in some type of cancer) but may also be important for immune response. Although the conclusions are interesting, the reviewer has few concerns in regards of the study, that would like to be addressed prior to publication:
- Figure 1 and 2 present qPCR data analysing the expression of HERV-K transcript (gag, env and pol) and interferon-related genes (IFNB1, ISG15 and IFIT1), respectively, in both patients and VERO cells infected with SARS-CoV-2. Although the data description in the result section is clear, the graph presentation is somehow counterintuitive. This is mostly due to the fact that ∆Ct is presented on the y-axis, rather than the ∆∆Ct or fold change within the infected patients/cells and the controls, which is the format commonly used for such data. That said, graph 2E is representing a significant decrease in IFIT1 when VERO cells are infected with SARS-CoV-2 with respect to the control (∆Ct higher in SARS-CoV-2 infected cells with respect to the control). This is somehow conflicting with the patient data (where for IFIT1 the ∆Ct is decreasing in infected patients – therefore increased gene expression) and the result description, stating: “In VERO cells, we found that the expression of the IFIT1 gene increased in VERO cells infected with SARS-CoV-2 (P<0.05). Could the authors clarify this point?
- Descriptions of both figure 1 and figure 2 are incorrect, since they refer to the wrong series of graphs: A to C are referring to COVID-19 patients and D to E/F are referring to VERO cells data, and not vice versa as described in the legend. Please correct accordingly.
- One of the analysis done by the group was the chromosomal localization of the highly expressed HERV-K (HML-2) gag, env and pol loci following Next Generation Sequencing analysis. According to the method section, the primer used for library preparation were the same as the ones used for the qPCR analysis. Previous genome wide analysis identified already several loci for gag, env and pol on the human chromosomes, and therefore it is not clear whether the loci defined as highly expressed in this study were the same as the previously identified ones, or a whole new pattern. Could the authors be more precise in these regards?
- Always regarding the highly expressed loci for gag, env and pol, it is not clear how the expression levels were identified. For example, was it done using the RPKM from the NGS sequencing data? A better description in the method section would be appreciated, as well as a table in the supplemental including the expression levels of the different loci with the defined unit used. Furthermore, a few sentences describing the characteristics of the three ORFs for the retroviral proteins Gag, Env and Pol, as well as how the different loci for each of them are identified would increase the smoothness of the whole manuscript and make it available for a broader reading audience.
- The authors performed Gene Ontology analyses feeding them with unigenes (42 from gag, 48 from env and 51 from pol), leading to identification of the most impacted biological processes, cell components and molecular functions. However, it is not clear how these unigenes were obtained, as well as which they are. A better description regarding how these genes were obtained in the method/result part, as well as a list (in the supplemental section would suffice) of such genes for each of the three HERV-K protein analyzed would clarify such question.
- Minor mistakes/typos:
- “Gene oncology” instead of “Gene ontology” in the method section
- “In VERO cells, we found that the expression of the IFIT1 gene increased in VERO cells infected…” results in a redundant sentence, please consider changing to make it a bit smoother.
Author Response
Point 1: This is somehow conflicting with the patient data (where for IFIT1 the ∆Ct is decreasing in infected patients – therefore increased gene expression) and the result description, stating: “In VERO cells, we found that the expression of the IFIT1 gene increased in VERO cells infected with SARS-CoV-2 (P<0.05). Could the authors clarify this point?
Response: Thank you so much for your careful check. This is indeed our mistake, and we are very sorry about this. And we have corrected it (line 186 on page 10). Thank you very much for your detailed review.
Point 2: Descriptions of both figure 1 and figure 2 are incorrect, since they refer to the wrong series of graphs: A to C are referring to COVID-19 patients and D to E/F are referring to VERO cells data, and not vice versa as described in the legend. Please correct accordingly.
Response: Thank you for your comments. We have examined carefully and there is no misrepresentation in figure 1 and figure 2. We wonder if there is any semantic difference. Can you tell me specifically?
Point 3: One of the analysis done by the group was the chromosomal localization of the highly expressed HERV-K (HML-2) gag, env and pol loci following Next Generation Sequencing analysis. According to the method section, the primer used for library preparation were the same as the ones used for the qPCR analysis. Previous genome wide analysis identified already several loci for gag, env and pol on the human chromosomes, and therefore it is not clear whether the loci defined as highly expressed in this study were the same as the previously identified ones, or a whole new pattern. Could the authors be more precise in these regards?
Response: Thank you for your rigorous consideration. We used (GRCh38/hg38) as the reference sequence, and compared the gene expression level of the HERV-K insertion loci in COVID-19 patients, and didn’t involve the new HERV-K insertion site. Therefore, the high expression loci defined in this study is the same as the previously identified ones. We have shown the primer-related loci in another article (PMID:35359739). If you are interested, you can take a look at figure 5A.
Point 4: Always regarding the highly expressed loci for gag, env and pol, it is not clear how the expression levels were identified. For example, was it done using the RPKM from the NGS sequencing data? A better description in the method section would be appreciated, as well as a table in the supplemental including the expression levels of the different loci with the defined unit used. Furthermore, a few sentences describing the characteristics of the three ORFs for the retroviral proteins Gag, Env and Pol, as well as how the different loci for each of them are identified would increase the smoothness of the whole manuscript and make it available for a broader reading audience.
Response: We gratefully thank you for your constructive remarks. As described by "Genome-wide distribution map" in “Materials and Methods”, we use the uniqueseqs values of different sequences in the second-generation sequencing data as the basis for judgment. The higher the uniqueseqs value, the more the sequence in cDNA, that is, the higher the transcriptional level. In this way, we calculate the percentage value (Appendix Table A1) of each sequence and visualize the percentage as color depth (Figure 3). In this paper, we mainly use specific primers to identify HERV-K gag, env, pol gene, which is not related to the ORF structure of HERV-K gag, env, pol gene. In order to make readers understand more smoothly, we have added the function of the HERV-K gag, env, pol gene in the preface. Please take a look at page 5, line 82.
Point 5: The authors performed Gene Ontology analyses feeding them with unigenes (42 from gag, 48 from env and 51 from pol), leading to identification of the most impacted biological processes, cell components and molecular functions. However, it is not clear how these unigenes were obtained, as well as which they are. A better description regarding how these genes were obtained in the method/result part, as well as a list (in the supplemental section would suffice) of such genes for each of the three HERV-K protein analyzed would clarify such question.
Response: Thank you very much for your valuable comments. We used the online website WebGestalt (http://www.webgestalt.org/) for GO analysis of genes at high expression loci. At the same time, we also include the specific information of unigenes in Appendix Table A2 for readers' reference.
- Minor mistakes/typos:
6.1 “Gene oncology” instead of “Gene ontology” in the method section
6.2 “In VERO cells, we found that the expression of the IFIT1 gene increased in VERO cells infected…” results in a redundant sentence, please consider changing to make it a bit smoother.
Response:
Thank you so much for your careful check. We have corrected the errors in the article. Please check them on page 8, line 147 and page 10, line 175. We are very grateful for your correction, which has made us make great progress.
Round 2
Reviewer 1 Report
The authors have been responsive to my previous critique. Thank you!
Reviewer 2 Report
The reviewer thinks that the author's answers suffice for the publication of the proposed manuscript, and has no further comments.